# Data-Driven Mathematical Model of Apoptosis Regulation in Memory Plasma Cells

**DOI:** 10.3390/cells11091547

**Published:** 2022-05-05

**Authors:** Philipp Burt, Rebecca Cornelis, Gustav Geißler, Stefanie Hahne, Andreas Radbruch, Hyun-Dong Chang, Kevin Thurley

**Affiliations:** 1German Rheumatism Research Center, 10117 Berlin, Germany; philipp.burt@drfz.de (P.B.); rebecca.cornelis@drfz.de (R.C.); gustav.geissler@gmail.com (G.G.); stefanie.hahne@drfz.de (S.H.); radbruch@drfz.de (A.R.); 2Institute for Theoretical Biology, Humboldt University, 10115 Berlin, Germany; 3Institute of Biotechnology, Department of Cytometry, Technische Universität, 10623 Berlin, Germany; 4Biomathematics Division, Institute of Experimental Oncology, University Hospital Bonn, 53127 Bonn, Germany

**Keywords:** plasma cells, mathematical model, apoptosis, immunological memory, cell signaling

## Abstract

Memory plasma cells constitutively produce copious amounts of antibodies, imposing a critical risk factor for autoimmune disease. We previously found that plasma cell survival requires secreted factors such as APRIL and direct contact to stromal cells, which act in concert to activate NF-κB- and PI3K-dependent signaling pathways to prevent cell death. However, the regulatory properties of the underlying biochemical network are confounded by the complexity of potential interaction and cross-regulation pathways. Here, based on flow-cytometric quantification of key signaling proteins in the presence or absence of the survival signals APRIL and contact to the stromal cell line ST2, we generated a quantitative model of plasma cell survival. Our model emphasizes the non-redundant nature of the two plasma cell survival signals APRIL and stromal cell contact, and highlights a requirement for differential regulation of individual caspases. The modeling approach allowed us to unify distinct data sets and derive a consistent picture of the intertwined signaling and apoptosis pathways regulating plasma cell survival.

## 1. Introduction

The vertebrate immune system has the unique ability to provide long-term protection against pathogens it has encountered previously. An important cellular correlate of this long-term immunity is the long-lived memory plasma cell, which constitutively secretes copious amounts of specific antibodies. Antibodies produced by memory plasma cells serve as a highly efficient barrier against re-infection with pathogens, but have also been associated with a variety of autoimmune diseases [1,2,3]. Memory plasma cell survival is conditional on signals provided to them in dedicated survival niches, organized by mesenchymal stromal cells, most prominently in the bone marrow, where memory plasma cells can survive for decades [1,3,4,5,6]. The survival of memory plasma cells is mediated through the integration of multiple input signals, including integrin-mediated contact to mesenchymal stromal cells and signaling via B-cell maturation antigen (BCMA) and Transmembrane activator and CAML interactor (TACI) [7,8]. In vivo, memory plasma cells can persist for a lifetime, i.e., they do not have a dedicated half-life: data indicate that plasma cells survive as long as they are provided with their niche [9,10,11].

Recently, we developed an in vitro culturing system for the study of plasma cell apoptosis [12]. We demonstrated that provision of A proliferation-inducing ligand (APRIL), addressing BMCA, and cell contact to the stromal cell line ST2, maintains primary bone marrow memory plasma cells in vitro for several days, thus preventing the rapid apoptosis process observed in absence of those factors [12]. Our in vitro system cannot yet reflect the very long in vivo survival times of plasma cells, which we think is mainly due to technical limitations—the co-cultured ST2 cells start overgrowing the plasma cells after ~5 days in culture. Nevertheless, we could show that APRIL and stromal cell contact act via different signaling pathways, namely, via nuclear factor kappa-light-chain-enhancer of activated B-cells (NF-κB) and phosphatidylinositol 3-kinase (PI3K), and address different cellular stress pathways, namely, endoplasmic reticulum stress and mitochondrial stress, respectively.

The wiring and control properties of the network at the interface of plasma cell death and extracellular factors such as APRIL and ST2 remain poorly understood. A prominent regulatory mechanism restricting the lifespan of plasma cells is the widely studied Bcl-2-associated X protein (BAX)-dependent apoptosis pathway [13], which ultimately results in the activation of caspases, a family of cysteine proteases. The complexity of the apoptotic network and the design principles leading to the apoptosis decision have been investigated extensively both by experiment and mathematical models [14,15], showing that caspase activation in single cells is rapid and irreversible [16,17], and that the apoptosis decision is well-described by a bi-stable and reversible switch [18,19,20,21]. However, it is less clear how different anti-apoptotic extracellular input signals affect the delicate machinery of cell death, and how well cell-type-specific aspects are captured by those modeling frameworks. 

Here, we developed a mathematical model of plasma cell survival in the bone marrow, using published [12,22] and yet unpublished data on key components of the plasma cell apoptosis pathway. To this end, we started from a core model based on established principles of the BAX-dependent apoptosis pathway, by adding the extracellular input signals APRIL and ST2 cells, and subsequently, by considering a more detailed interaction network of several caspase proteins downstream of the BAX-module. We found that the survival factors APRIL and ST2 cells have differential rather than additive roles in the regulation of plasma cell lifespan extension, acting in different ways and primarily on different parts of the network. Further, our analysis underlines the essential aspect of differential caspase regulation for the apoptosis decision and provides insight into the parameters that could be manipulated to alter plasma cell lifespan.

## 2. Materials and Methods

### 2.1. Cell Culture and Flow-Cytometric Measurements of Apoptosis Proteins

Experiments were performed with the same cell culture system as previously reported [12]. In brief, mice were primed with 100 µg 4-hydroxy-3-nitropheylacetyl hapten-coupled chicken gamma globulin (NP-CGG) in incomplete Freud‘s Adjuvants (IFA) intraperitoneally (i.p.). Mice were challenged twice after the prime with the same injection in cycles of 21 days. Plasma cells were magnetically isolated from immunized mice more than 30 days after the 2nd boost using a two-step protocol, including depletion of B220- and CD49b-expressing cells and subsequent positive enrichment of CD138^high^ PCs. Isolated long-lived PCs from the bone marrow were cultured in RPMI1640 medium supplemented with 10% FCS, 100 U/mL Penicillin, 100 mg/mL streptomycin, 0.1% b-Mercaptoethanol, 25 mM HEPES buffer, and with/out 50 ng/mL multimeric APRIL. Cultures were kept under physiological oxygen levels in a hypoxia chamber with 4.2% O_2_ and 5% CO_2_ at 37 °C. The medium was changed at day 3 of culture. For the co-culture, 2500 ST2 cells were seeded in a 96-well plate one day before memory plasma cell isolation. Memory plasma cells were plated on top of the ST2 cell layer in a 1:1 ratio (5000 PC and 5000 ST2 cells). For analysis, cells were either fixed with PFA or stained directly and scraped off the plate before measurement. 

Intracellular antigens were stained by fixing cells with PFA and permeabilization with methanol. To prevent unspecific binding, cells were pre-incubated with blocking buffer and subsequently stained with primary antibody for 1 h and, if necessary, with secondary antibody for 30 min (cf. Table 1). Samples were analyzed using a MacsQuant analyzer and FlowJo software. The following antibodies were used in the experiments:

### 2.2. Data Analysis and Statistics

To estimate the half-life for plasma cells under different conditions, we fitted an exponential decay function $f\left(t\right)=100e^{-\lambda t}$ to each individual time series. The resulting decay rates were converted to half-lives for each condition according to: $t_{1/2}=log\left(2\right)/\lambda$, where λ represents the average decay rate summarized from the individual fit procedures. To compare average protein concentrations, geometric means for each protein measured under a specific condition were first normalized to the respective medium condition and then compared using an unpaired Student’s *t*-test. Uncertainties of protein ratios were calculated from the normalized protein concentrations by bootstrapping. To this end, we repeatedly drew from the original samples with replacement to estimate mean and standard deviation of the data.

### 2.3. Mathematical Models and Numerical Simulations

A complete description of the model equations is available as Appendix A. In brief, the model consists of two parts:

***(a) BAX-dependent apoptosis:*** We consider (i) production and degradation of MCL-2 family proteins in dependence of the input stimuli APRIL and ST2, (ii) complex formation between pro- and anti-apoptotic proteins, and (iii) a dependence of the average half-life of the plasma cell population on the average concentration of activated BAX (BAX*) as follows:(1)$$\lambda =\gamma \frac{{\left[BAX^{*}\right]}^{3}}{K_{Bax}^{3}+{\left[BAX^{*}\right]}^{3}}.$$

***(b) BAX-independent regulation of caspases:*** To consider the effect of direct caspase regulation by APRIL and ST2 on the apoptosis decision, we extended the BAX-Apoptosis model and assumed additional regulation of the cell-death rate *λ* as follows:(2)$$\;\lambda =\left[\gamma \left(\frac{{\left[BAX\right]}^{3}}{K^{3}{}_{Bax}+{\left[BAX\right]}^{3}}+\frac{\kappa }{1+\alpha f\left(\left[APRIL\right]\right)}\right)\left(\frac{1}{1+\beta f\left(\left[ST2\right]\right)}\right)\right].$$

That means that the cell-death rate γ is increased due to the activity of caspase 3/7, which is induced through a combination of BAX-dependent and independent effects. Those BAX-independent effects stem from activity of caspase 12, which can be inhibited by APRIL. Finally, the activity of caspase 3/7 can be inhibited by ST2. Here, the fitting parameter κ is the relative effect of caspase 12 activity, the fitting parameter *β* denotes the maximal inhibitory effect ST2 on caspase 3/7, and we set α=10 for the inhibitory effect size of APRIL. For model fitting to our data in absence and presence of APRIL and ST2, we adopt a Boolean formulation for the regulatory function, $f\left(x\right)=\{\begin{array}{c} 1,\;x>0\\ 0,\;x=0 \end{array}$. Modification of this function into a more specific Hill-type form such as $f\left(x\right)=x^{n}/\left(x^{n}+K\right)$ is straight-forward.

All model simulations were carried out in Python 3.8. Ordinary differential equations were solved using the scipy.odeint routine. For curve-fitting, least-squares optimization was employed using the Levenberg–Marquardt algorithm implemented in the Python lmfit library with the cost function $\chi^{2}=\sum_{i=1}^{N}{\left(\frac{y_{i}-f\left(x_{i}\right)}{\sigma_{i}}\right)}^{2}$, where $f\left(x_{i}\right)$ represents model predictions, $y_{i},\sigma_{i}$ represent measured data and data uncertainties, respectively, and *N* denotes the number of data points. 

For the calculation of AIC, we used the convention $AIC=2k+N\;ln\left(\chi^{2}/N\right)$, where *k* represents the number of fit parameters. For the perturbation analysis, we defined the effect size as the log2-fold change in half-life (or *BAX**) between a model simulation with best-fit parameter values and the model simulation with perturbed parameters. The parameters were either up- or downregulated by 2-, 5-, or 10-fold variation of the best-fit parameter value.

## 3. Results

### 3.1. A Quantitative Model of BAX-Dependent Apoptosis in Plasma Cells

Previous experiments have shown that plasma cells are not intrinsically long-lived but that provision of the cytokine APRIL combined with cell–cell contact, conferred by the stromal cell line ST2, keeps them alive in vitro for several days. Each factor alone proved to be insufficient to prevent rapid apoptosis [12] (Figure 1A). Re-plotting the data from Cornelis et al. [12] separately for individual experiments revealed an exponential survival curve for cultured plasma cells, where the half-life is significantly increased from less than 1 day to more than 6 days after addition of APRIL and ST2 (Figure 1B,C and Appendix A). How these signals integrate and affect the underlying complex regulatory network facilitating apoptosis decisions remains unclear (Figure 1A). In particular, we wondered whether the experimental observations could be reconciled with an additive model, where APRIL and ST2 synergistically act to inhibit BAX-dependent apoptosis, or alternatively, whether APRIL and ST2 address fundamentally distinct parts of the apoptosis network.

In plasma cells, APRIL stimulates the NF-κB signaling pathway [12,23,24,25] and ST2 acts via the PI3K pathway to stimulate the transcription factor Forkhead-Box-Protein O1/3 (FoxO1/3), as shown by a series of targeted inhibition experiments [12,26]. Therefore, we started our analysis based on the working hypothesis that ST2 and APRIL act by targeting components of the BAX-dependent apoptosis pathway. In brief, oligomerization of two proteins, BAX and BAK (BCL-2-antagonist/killer 1), that are localized at the mitochondrial outer membrane and cytosol, leads to the formation of the apoptotic pore. Pore formation precedes cytochrome c and apoptosis-inducing factor release into the cytosol and downstream activation of caspases. The BCL-2 family comprises a group of proteins that is critical for the control of apoptosis by regulating the oligomerization of BAX and BAK. Localized mainly at the mitochondria, they can be divided into pro- and anti-apoptotic proteins. The anti-apoptotic proteins (B-cell lymphoma 2, BCL-2 and myeloid cell leukemia 1, MCL-1) prevent apoptotic pore formation and preserve the integrity of the mitochondrial membrane by binding to BAX and BAK. Pro-apoptotic proteins (Bcl-2-like protein 11, BIM, and NOXA), next to BAX and BAK, compete in binding to the anti-apoptotic proteins, thereby neutralizing them. Thus, the ratio of pro- to anti-apoptotic proteins plays a decisive role in apoptosis regulation [27]. The expression of the anti-apoptotic proteins BCL-2 and MCL-1 was shown to be upregulated in plasma cells, but while BCL-2 seems to be dispensable for the maintenance of memory plasma cells, MCL-1 is essential for their survival [28]. In multiple myeloma, MCL-1 strongly binds to BIM, thereby blocking apoptosis [29]. In addition to BIM, MCL-1 was reported to interact with NOXA [30].

To derive a mathematical model capturing those processes specifically for memory plasma cells, we measured the abundance of several key components of the BCL-2 family, namely, BIM, BCL-2, NOXA, and MCL-1, after 3 days in our established in vitro culture system [12] with and without stimulation by APRIL and ST2, by immune-fluorescent single-cell staining (Figure 2A). We found that ST2 cells had a significant negative effect on BIM, NOXA, and MCL-1 concentrations, whereas for APRIL alone, no significant effects could be detected. The divergent effects of APRIL and ST2 cells on different members of the BCL-2 family were far more pronounced when considering the ratios within pairs of pro- and anti-apoptotic proteins (Figure 2B). Taking that into account, we also considered upregulation of BCL-2 via APRIL stimulation as a plausible mechanism, although it was not significant at direct comparison (Figure 2A).

We used the information gained by experiments to establish a refined regulatory network of BAX-dependent plasma cell apoptosis (Figure 2C). Combined with a recently published compilation of quantitative data for the BCL-2 interactome (Table 2) [22,31,32,33,34], we were now in a position to formulate and annotate a specific mathematical model of BAX-dependent apoptosis in plasma cells (Methods and Appendix A). The only parameters lacking good estimates from the literature are the values describing protein production rates, which we therefore used as fitting parameters. Indeed, the resulting model was able to describe the new data set of BCL-2 family member protein abundance (Figure 2D). 

Having established a mathematical formulation of the BAX-dependent apoptosis pathway in the context of plasma cell survival, we proceeded to the original question of whether APRIL- and ST2-dependent regulation of that pathway are sufficient to explain the observed survival kinetics shown in Figure 1B. Interestingly, we found that our model was well able to fit the protein and survival data individually (Appendix A), but it was not able to capture both protein and survival data at the same time within the expected range of the experimental data (Figure 2E). In fact, the annotated model predicted a positive regulatory effect of ST2 on the average level of activated BAX (BAX*) in the plasma cell population (Figure 2F). That ST2-dependent elevation of BAX* cannot be fully compensated for by APRIL, which explains the disconnect of that data-derived model parameterization with our cell-survival data (Figure 2E). Hence, we concluded that APRIL- and ST2-driven regulation of BAX-dependent apoptosis alone is insufficient to explain the observed survival curves, and rather, additional regulatory mechanisms must be considered. 

### 3.2. Direct Regulation of Caspase Proteins Is Required for Effective Control

Caspases are not only mediators, but also critical regulators of apoptosis [35,36], and our recent data suggest that APRIL and ST2 have different roles in their regulation [12]. Specifically, we found that ST2-induced PI3K inhibits caspases 3 and 7, whereas APRIL-induced NF-κB inhibits the ER-stress-induced apoptosis mediated via caspase 12. Therefore, we supplemented our core model of BAX-dependent apoptosis by considering a network of caspase regulation (Figure 3A) (Methods). To specify the model, we added the two new fitting parameters describing the extended caspase network, and we found that this extended model indeed captures the individual and combined effects of APRIL and ST2 (Figure 3B,C). In particular, in contrast to the BAX core model, we obtained good results fitting the new caspase-related parameters to the plasma cell survival data (Figure 3C).

We could previously show that targeted inhibition of caspases had pronounced effects on plasma cell survival [12]. The successful data annotation gave us the opportunity to quantify the model dependency on these signaling pathways, as well as the effects of BCL-2-family members (Figure 3D and Appendix A). As expected, the analysis showed strong effects for the regulation of caspase 3/7. That is in agreement with our previous data and with the intuition gained from Figure 2E, that direct regulation of caspase 3/7 through the ST2/PI3K pathway is required to counteract the pro-apoptotic role of ST2 within the BAX-dependent apoptosis pathway. Furthermore, our previous experiments have revealed quantitative effects in response to partial inhibition of PI3K and NF-κB [12]. Indeed, model simulations indicated a gradual response to PI3K and NF-κB, in contrast to a rather all-or-none type of regulation in the case of apoptotic proteins. Regulation of NF-κB turned out to have the largest effect on plasma cell lifespan (Figure 3E), also in line with our previous experiments.

Taken together, our analysis suggests that a unified description of the two considered data sets requires consideration of both BAX-dependent and BAX-independent regulation of caspases via both APRIL/NF-κB and ST2/PI3K.

### 3.3. Full Model Topology Is Essential to Describe Plasma Cell Survival

To further test whether our proposed model topology was necessary to describe all available data, we derived a set of sub-models lacking one or more components of the full model (Figure 4A). For each possible combination, we fitted the model to the available data and compared the fitting result to the original model fit (Figure 4B and Appendix A). The full model had clearly the smallest fitting error (*χ*^2^), and models lacking regulatory effects of either APRIL, ST2, or both on caspase activity all provided similar fitting quality (Figure 4B). For a refined model comparison, we employed Akaike’s information criterion (AIC) (Figure 4C), a well-established metric for comparing models with differing numbers of fitting parameters [37]. AIC values lack a direct interpretation, and thus we only consider differences between models (−∆AIC), and here we show −∆AIC in relation to the model with poorest fit quality (Figure 4C). A difference of ∆AIC > 2 is usually regarded as significant, and therefore, our analysis clearly rules out all sub-models (Figure 4C). Nevertheless, the ∆AIC representation can also be regarded as a ranking of the most critical model components for explanation of the available data. As such, it is intriguing that in our model, ST2-driven regulation of caspase 3/7 is the most critical individual factor (panel (iv) in Figure 4A,C), even exceeding the effect of by-passing caspase regulation completely (panel (vi)). Hence, our model analysis supports both a strong role for BAX-independent caspase regulation and the critical need of ST2-derived signals for effective regulation of plasma cell survival.

## 4. Discussion

In this work, we developed a mathematical model describing the survival of plasma cells in the bone marrow. We used the model to unify two acquired data sets, namely, abundance of pro- and anti-apoptotic proteins and plasma cell survival kinetics in the presence of the soluble factor APRIL and/or stromal ST2 cells as cell-contact-dependent survival signal. In the annotated model, the balance of survival proteins is a critical factor for the lifespan of plasma cells. Hence, we found that a combination of direct caspase regulation together with regulation of BAX-dependent pathways is essential to explain all available data. Furthermore, perturbation analysis of the model largely recovered previously established [12] effects of targeted inhibition experiments of caspases as well as NF-κB/PI3K signaling. 

A characteristic property of plasma cells is the contribution of the endoplasmic reticulum (ER) to apoptosis regulation [13,38]. Since plasma cells as the primary antibody-secreting cell type produce vast amounts of proteins, they experience high ER-stress due to protein miss-folding. In plasma cells, BAX was not only found in the mitochondrial membrane but also localized at the ER [13]. We could recently show that the plasma cell microenvironment, composed of stromal cell contact and the cytokine APRIL, counteracts the activation of caspases 3 and 7 and caspase 12, respectively [12]. Caspases 3 and 7, that are activated upon mitochondrial stress [35], are regulated by PI3K signaling upon cell contact to stromal cells. Activation of the ER-associated caspase 12 is inhibited by APRIL signaling via the NF-κB pathway [12]. Here, our model simulations further supported that finding of differential regulation of caspases by APRIL and ST2, since without such additional flexibility the model was not able to explain both the apoptosis kinetics and the protein abundance data. For therapy, targeted depletion of memory plasma cells has been proposed as a promising strategy in autoimmune diseases such as lupus erythematosus [2]. On the other hand, memory plasma cells can provide long-lasting protection against infections, as recently demonstrated for mild SARS-Cov2 infections in humans [39]. The lifespan of immune cell populations is tightly controlled and is a critical property in a range of inflammatory processes, as shown previously in the context of selective expansion of lymphocyte subtypes [40,41] and NK cell subpopulations [42], amongst others. Therefore, a better quantitative understanding of the regulatory processes controlling the lifespan of memory plasma cells and other immune cell types will be critical for optimized targeted therapy and personalized medicine approaches.

The regulatory networks driving T-cell and B-cell differentiation and transition to memory cell types have been widely studied by combined computational and mathematical modeling studies. Such efforts are still rare in the analysis of immunologic cell death decisions and specifically in plasma cell biology. A recent study focused on the generation and composition of the survival niche [43], and a previous review estimated the long-term development of the pool of available memory plasma cells [44]. Here, using cultured plasma cells as a model system, we developed a fully annotated mathematical model of apoptosis regulation in memory plasma cells. This work may set the stage for further quantitative and mechanistic analyses of the differential regulation of apoptosis pathways in specific tissue and immune cell types.

## Figures and Tables

**Figure 1 cells-11-01547-f001:**
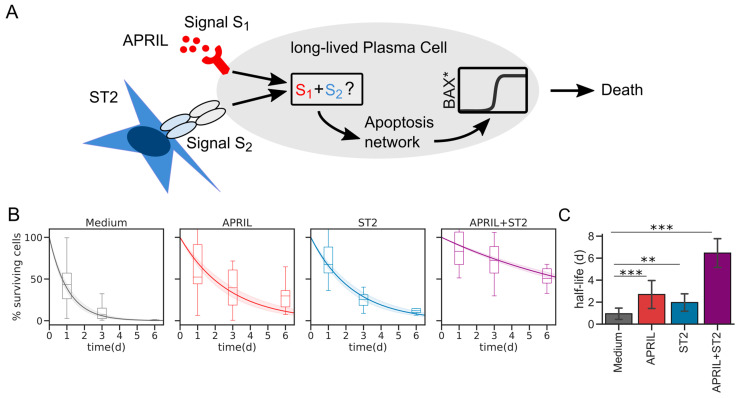
Regulation of plasma cell survival via APRIL and ST2. (**A**) Schematic signal integration of ST2 and APRIL. How the signals are integrated into the BAX-dependent apoptosis network is yet unclear. Time course data were taken from Ref. [12] and are here represented as Box plots. Data were fitted to exponential curves, each experiment at a time (Appendix A). Curves shown here represent mean + s.e.m. of those fit results. BAX* denotes activated BAX. (**C**) Average half-lives of each condition based on the fitting procedure from (**B**). ** *p* < 0.01, *** *p* < 0.001, n > 12 fits per condition. Error bars represent standard deviation.

**Figure 2 cells-11-01547-f002:**
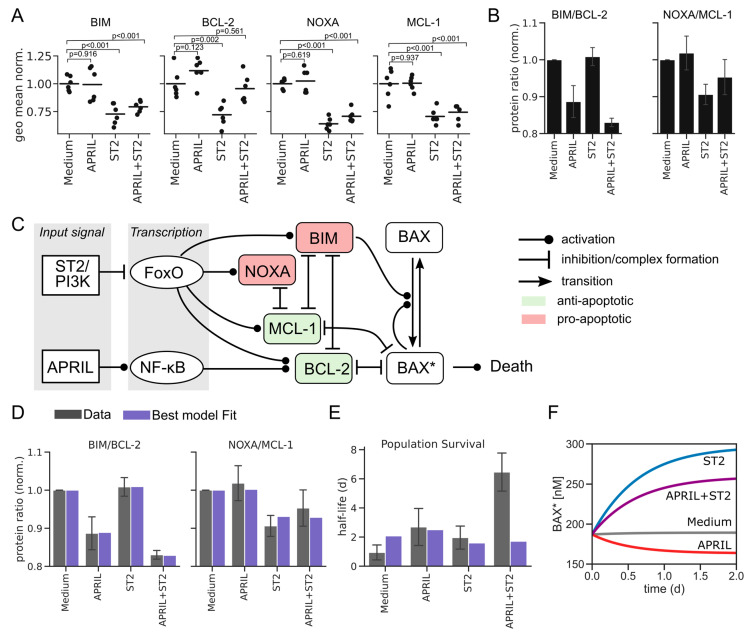
A mechanistic model quantifies the BAX-dependent apoptosis network in plasma cells. (**A**) Geometric mean of measured core proteins in APRIL/ST2/APRIL + ST2 environments normalized to the respective concentration without stimulus (Medium) (n = 6 biological replicates). (**B**) Ratios of indicated proteins taken from (**A**) after normalization to Medium condition. (**C**) Model scheme. APRIL- and ST2-induced PI3K signal via NF-κB and FoxO, thereby affecting BCL-2 family protein abundance data, as shown in panel A. The model includes pro-apoptotic proteins BIM and NOXA (red) and anti-apoptotic proteins BCL-2 and MCL-1 (green). BAX* denotes activated BAX. (**D**) Protein ratios (see A) fitted to the model shown in panel B ($\chi^{2}=0.001$). Error bars represent standard deviation. (**E**) Model fit to survival kinetics (Figure 1B) after fitting the protein ratios ($\chi^{2}=27.12$). (**F**) Time-course simulation of the mechanistic apoptosis model for different inputs as indicated.

**Figure 3 cells-11-01547-f003:**
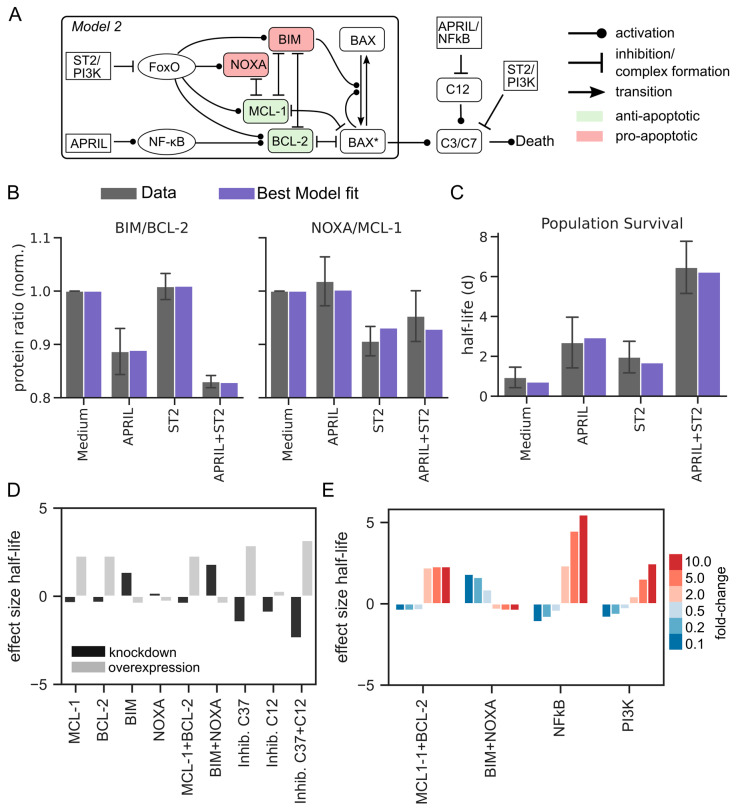
Unifying the data sets on MCL-2 family members and plasma cell survival data requires direct regulation of caspases. (**A**) Model scheme, combination of the mechanistic mitochondrial apoptosis model (see Figure 2B) with ER-stress-induced caspase activation. (**B**,**C**) Combined model fit to protein data and survival kinetics ($\chi^{2}=0.152$). Error bars represent standard deviation. (**D**) Effect on half-life for simulated partial inhibition or overexpression of BAX/BCL-2-family proteins and caspases 3, 7, and 12. Parameters were varied by one order of magnitude with respect to the best-fit parameter value. (**E**) Effect of quantitative inhibition or overexpression of NF-κB, PI3K, and apoptotic regulators on half-life. Parameters were varied as 2-, 5-, or 10-fold change with respect to the best-fit parameter value.

**Figure 4 cells-11-01547-f004:**
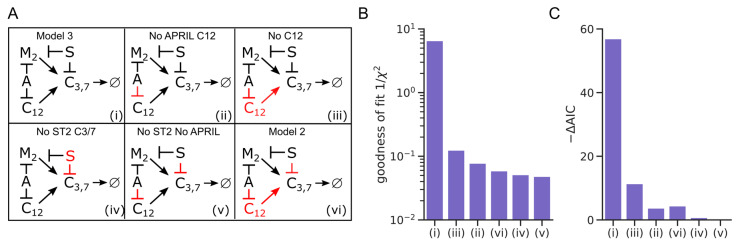
All considered regulatory processes are required to explain the data. (**A**) Different hypothetic network topologies (ii–vi) as submodules of the model shown in (3A), here presented as a condensed version (i). Abbreviations: M2: Model 2; A: APRIL; S: ST2; C12: Caspase 12; C3/7: Caspase 3, Caspase 7. Model components removed from the full model are shown in red. (**B**) goodness-of-fit (1/root-mean-squared error)) for each submodel (ii–vi). (**C**) Akaike information criterion (−ΔAIC) for each sub-model compared to the full model (i), where the model with smallest −ΔAIC is used as a reference and set to 0.

**Table 1 cells-11-01547-t001:** Antibodies used for flow-cytometric measurement of apoptosis proteins.

Antibody Clone	Manufacturer	Catalog nr.
Anti-mouse BCL-2, REA356	Miltenyi Biotec	Catalog # 130-105-474
Anti-mouse BIM, 14A8	Milipore	Catalog # MAB17001
Anti-mouse CD138, REA104	Miltenyi Biotec	Catalog # 130-102-318
Anti-mouse MCL-1, Y37	Abcam	Catalog # ab32087
Anti-mouse NOXA, 114C307	Abcam	Catalog # ab13654

**Table 2 cells-11-01547-t002:** Parameter values used in the mathematical models.

Parameter	Value	Unit	Role	Source
a_BCL-2_	0.11	-	Effect APRIL on g_BCL-2_	Fit
s_MCL-1_	0.37	-	Effect ST2 on g_MCL-1_	Fit
s_BCL-2_	0.53	-	Effect ST2 on g_BCL-2_	Fit
s_BIM_	0.49	-	Effect ST2 on g_BIM_	Fit
s_NOXA_	0.40	-	Effect ST2 on g_NOXA_	Fit
γ	0.43	d^−1^	Max. effect size BAX*	Fit
κ	1.78	-	Max. effect Caspase 12	Fit
β	2.65	-	Inhibition strength ST2 on Caspase 3,7	Fit
d_MCL-1_	16.4	d^−1^	Decay rate MCL-1	[31]
d_BCL-2_	0.86	d^−1^	Decay rate BCL-2	[31]
d_BIM_	5.94	d^−1^	Decay rate BIM	[33]
d_NOXA_	32.8	d^−1^	Decay rate NOXA	[34]
d_BAX_	1.38	d^−1^	Decay rate BAX	[32]
K_d,3_	2.0	nM	Dissociation constant	[22]
K_d,4_	22.0	nM	Dissociation constant	[22]
K_d,5_	40.0	nM	Dissociation constant	[22]
K_d,6_	2.50	nM	Dissociation constant	[22]
K_d,7_	68.0	nM	Dissociation constant	[22]
k^+^	0.17	µM d^−1^	Complex association rate	-
k_1_	43.2	µM d^−1^	BAX activation rate	-
k_2_	8.64	µM d^−1^	BAX* deactivation rate	-
g_p,0_	0.86	µM d^−1^	Basal protein growth	-
α	10.0	-	Inhibition strength APRIL on Caspase 12	-
K_BAX_	200	nM	Half-saturation constant BAX-half-life	-

## Data Availability

Data are contained within the article or Appendix A.

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
