# Peer review of "Data-Driven Mathematical Model of Apoptosis Regulation in Memory Plasma Cells"

_cells, 2022, doi:10.3390/cells11091547_

Round 1

Reviewer 1 Report

I have gone through the manuscript. Topic seems intriguing but it needs to be revised. Targeted inhibition of FOXO and NFKB as well as reconstitution of FOXO and NFKB will add more concrete evidence to this signaling pathway. Why authors only analyzed caspase caspase3/7 and caspase 12? Do the authors have previous evidence of any role of caspase 9 in this pathway?

Author Response

Please find the point-by-point response below, and the the revised manuscript in correction mode attached.

I have gone through the manuscript. Topic seems intriguing but it needs to be revised. Targeted inhibition of FOXO and NFKB as well as reconstitution of FOXO and NFKB will add more concrete evidence to this signaling pathway.

We thank the reviewer for his/her effort and for the overall positive evaluation. Further, we thank the reviewer for this comment, which made us realize that we did not point out the connection of the model derived in this manuscript to experiments performed in previous work with enough clarity.

Indeed, the targeted inhibition of FOXO and NFKB in plasma cells has been performed and is shown in Cornelis et al., Cell Reports 2020, Figure 3C and Figure S2B, respectively. Downregulation of FOXO following PI3K activation is a critical step required for the survival of plasma cells, as a consequence of contact to stromal cells. In accordance, siRNA-mediated targeted inhibition of FOXO abrogated the need for stromal cell contact and resulted in the survival of plasma cells without stromal cells. The established effects of FOXO on plasma cell survival motivated the design of the proposed mathematical model with FOXO being the primary target of PI3K-mediated regulation of the BAX/BCL-2 apoptosis network (see Figure 2C). On the other hand, activation of NFkB by binding of APRIL to its receptor, and activation of PI3K upon stromal cell contact, are also required for plasma cell survival. We previously found that targeted blockade of NFkB with IKK16 results in early (d1-3) death of plasma cells, and we found similar results for targeted inhibition of PI3K by Wortmanin and LY294002 (Cornelis et al., Figure 2A).

To further analyze the effects of individual regulators in our model in a quantitative fashion, we added a new model analysis to the manuscript (Figure 3E) that highlights the gradual effects for targeted inhibition of NFkB and PI3K, both of which are in good agreement with the previously obtained data. Moreover, we rewrote the corresponding paragraph in the text (line 197), and we now highlight the connection to the previous experiments at several other instances in the text, for instance lines 48, 99, 177, 249.

We agree with the reviewer that reconstitution of FOXO and NFkB would add additional validation on the signaling pathways described in this manuscript. However, we think that reconstitution experiments would prove very challenging and would themselves have several drawbacks. Our expectation would be that reconstitution of FOXO levels would immediately kill the plasma cells, making further analysis of signaling pathways impossible. NFkB requires activation for its activity as transcription factor, i.e. addition of survival signals such as APRIL will be necessary. In these conditions, we could not preclude that APRIL also addresses other signaling pathways. Further, reconstitution with a constitutively active form of NFkB could result in many unwanted or interfering effects due to the pleiotropic nature of NFkB. Therefore, we decided already in the previous paper to focus on inhibition rather than reconstitution experiments.

Why authors only analyzed caspase caspase3/7 and caspase 12? Do the authors have previous evidence of any role of caspase 9 in this pathway?

We have focused our analysis on caspases 3/7, as they represent the executioner caspases. Caspase 12 has previously been shown to play a central role in plasma cell death and was shown to be downstream of endoplasmic reticulum stress. We realize that measuring of caspase 9 activity would bring about additional evidence for mitochondria-mediated cell death, and we have attempted the analysis of caspase 9 activation. However, due to either technical or biological reasons, we could not achieve a reliable and controlled analysis of caspase 9 activation in plasma cells. As we see differential inhibition of the activation of caspases 3/7 and caspase 12, we think that the data still support our model which suggests that integrin-mediated stromal cell contact and APRIL signaling provide resilience to two major sources of cell stress in plasma cells, namely mitochondrial stress and endoplasmatic reticulum stress, respectively.

Reviewer 2 Report

The authors performed experiments with ex vivo plasma cells isolated from mice after two-shot immunization, and study the factors that mediate plasma cell survival with an in vitro system. The in vitro system cultivates the plasma cells on a layer of stromal cells as feeding layer, which allows to study the contribution of two main signals on plasma cell survival.

They show that both ST2 and april are necessary to maintain cells alive for around one week, and work in a synergistic manner.

Based on protein measurement of main protein factors of the apoptosis pathway by flow cytometry at one time-point (day 3), the authors propose two ODE models.

With a simple ODE model using known signaling pathways for ST2 and APRIL and the main caspase apoptosis pathway, it is not possible to explain the synergistic effect of these two factors. Of note, the authors used many parameters from literature during fitting, leaving only a few parameters to fit, which is suited to the small dimension of the data (one time-point). These fixed parameters were likely crititical to find that the simple model is not consistent with the data.

Then, they proposed a more complex model including an additional layer of regulation in the caspace system. Not only this model explained the data well, but removing any pathway led to a drop in fitting quality (measured as delta AIC).

The article is well written, combines new experimental data and extracts, I believe, the maximum information possible from literature and their own data. Understanding memory cell survival in vivo is critical to understand long-term vaccine protection, and therefore this model is welcome to the community as to further modulate plasma cell survival.

Small details:

  • can the authors write if the medium changed during the 7 days?
  • I believe the data are biological replicates, not technical, please write it down
  • can the authors give the fitting cost of each model and write down the formula of their cost function, including the formula for the AIC they have chosen (there are some variations in the field)

Author Response

Please find the point-by-point response below, and the revised manuscript attached.

The authors performed experiments with ex vivo plasma cells isolated from mice after two-shot immunization, and study the factors that mediate plasma cell survival with an in vitro system. The in vitro system cultivates the plasma cells on a layer of stromal cells as feeding layer, which allows to study the contribution of two main signals on plasma cell survival. They show that both ST2 and april are necessary to maintain cells alive for around one week, and work in a synergistic manner. Based on protein measurement of main protein factors of the apoptosis pathway by flow cytometry at one time-point (day 3), the authors propose two ODE models.

With a simple ODE model using known signaling pathways for ST2 and APRIL and the main caspase apoptosis pathway, it is not possible to explain the synergistic effect of these two factors. Of note, the authors used many parameters from literature during fitting, leaving only a few parameters to fit, which is suited to the small dimension of the data (one time-point). These fixed parameters were likely critical to find that the simple model is not consistent with the data. Then, they proposed a more complex model including an additional layer of regulation in the caspase system. Not only this model explained the data well, but removing any pathway led to a drop in fitting quality (measured as delta AIC). The article is well written, combines new experimental data and extracts, I believe, the maximum information possible from literature and their own data. Understanding memory cell survival in vivo is critical to understand long-term vaccine protection, and therefore this
model is welcome to the community as to further modulate plasma cell survival.

We thank the reviewer for these positive comments.

Small details:

Can the authors write if the medium changed during the 7 days?

Indeed, the medium was changed at day 3 of culture. We have added this information to the revised manuscript.

I believe the data are biological replicates, not technical, please write it down.

Correct, thanks for the notice. We added that information in the corresponding caption, Figure 2A.

Can the authors give the fitting cost of each model and write down the formula of their cost function, including the formula for the AIC they have chosen (there are some variations in the field).

We added the formulas used to derive fitting and AIC metrics in the Section 3, Materials and Methods, and we now provide fitting cost values χ2 in the respective figure captions.

Round 2

Reviewer 1 Report

The authors have addressed all concerns properly.